# The Nuclear Pore Complex: Birth, Life, and Death of a Cellular Behemoth

**DOI:** 10.3390/cells11091456

**Published:** 2022-04-25

**Authors:** Elisa Dultz, Matthias Wojtynek, Ohad Medalia, Evgeny Onischenko

**Affiliations:** 1Institute of Biochemistry, Department of Biology, ETHZ Zurich, 8093 Zurich, Switzerland; m.wojtynek@bioc.uzh.ch; 2Department of Biochemistry, University of Zurich, 8057 Zurich, Switzerland; omedalia@bioc.uzh.ch; 3Department of Biological Sciences, University of Bergen, 5020 Bergen, Norway

**Keywords:** nuclear pore complex, nucleoporin, NPC, membrane fusion, Ran, lipids, assembly factor, amphipathic helix, nuclear transport receptor, FG repeats, Brl1, autophagy, ageing, aggregation, neurodegneration

## Abstract

Nuclear pore complexes (NPCs) are the only transport channels that cross the nuclear envelope. Constructed from ~500–1000 nucleoporin proteins each, they are among the largest macromolecular assemblies in eukaryotic cells. Thanks to advances in structural analysis approaches, the construction principles and architecture of the NPC have recently been revealed at submolecular resolution. Although the overall structure and inventory of nucleoporins are conserved, NPCs exhibit significant compositional and functional plasticity even within single cells and surprising variability in their assembly pathways. Once assembled, NPCs remain seemingly unexchangeable in post-mitotic cells. There are a number of as yet unresolved questions about how the versatility of NPC assembly and composition is established, how cells monitor the functional state of NPCs or how they could be renewed. Here, we review current progress in our understanding of the key aspects of NPC architecture and lifecycle.

## 1. Introduction

The central hallmark and name-giving feature of all eukaryotic cells is the nucleus (from the Greek “karyon” meaning “kernel”). This organelle compartmentalizes the genetic information within a double lipid membrane bilayer called the nuclear envelope (NE), thus separating transcription and translation into different subcellular locations. In other membrane-bound organelles, selective transport of ions, metabolites and other substrates is facilitated by a large number of different transmembrane channels. Remarkably, all transport across the NE is mediated by a single versatile channel that fulfills the challenge of selectively importing and exporting a myriad of different cargoes: the nuclear pore complex (NPC). The NPC is one of the largest protein complexes in eukaryotic cells, consisting of more than 500 individual proteins in *Saccharomyces cerevisiae* and over 1000 proteins in human cells. These proteins, known as nucleoporins (NUPs), are the basic building blocks of the NPC. In this review, we describe the current view on the architectural concepts of the NPC and the stages of its life from assembly to decay. For simplicity, we will use the budding yeast (*S. cerevisiae*) nomenclature for NUPs and complexes if not specified otherwise.

## 2. Tour of the Nuclear Pore Complex Architecture

Depending on the species, the NPC has an outer diameter of ~120–130 nm and a height of 50–80 nm [1,2,3,4,5,6,7]. The core of the NPC has an eightfold rotational symmetry around the nucleocytoplasmic axis and can be described as a three-ring assembly: an inner ring in the plane of the NE, sandwiched by outer rings on the cytoplasmic and nucleoplasmic sides (Figure 1). Although most of the structured core of the NPC is symmetric, the cytoplasmic and nucleoplasmic rings carry specialized attachments: the cytoplasm-facing mRNA export platform and the fishtrap-like nuclear basket [8,9] (Figure 1). The central channel of the NPC has a diameter of ~40–60 nm [1,2,3,4,5,6,7] and is filled by intrinsically disordered domains rich in phenylalanine-glycine (FG) repeats, which are present in a third of NUPs and make up 9 MDa of the ~50 MDa mass of the *S. cerevisiae* NPC [10]. Although the exact make-up of the permeability barrier established by these domains remains unclear (reviewed in [11]), it allows biomolecules of less than ~40 kDa to freely diffuse through the NPC, whereas larger cargoes rely on a sophisticated nucleocytoplasmic transport machinery involving nuclear transport receptors (NTRs) and fueled by a gradient of the small GTPase Ran (reviewed in [12]).

### 2.1. Inner Ring: The Flexible Core of the Nuclear Pore Complex

The architecture of the inner ring, with its symmetry along the nucleocytoplasmic axis, is highly conserved [13]. It coats the NE with eight spokes positioned around the central transport channel, each formed by three layers. Closest to the central channel, the innermost layer consists of the so-called channel nucleoporin heterotrimer (Nup49, Nup57, Nsp1), which projects intrinsically disordered FG-rich segments into the central NPC channel [14] (Figure 1). The outer, membrane-binding layer is composed of the α-solenoid/β-propeller domain paralogues Nup157/Nup170, which bind the NE via an amphipathic lipid packing sensor (ALPS) motif positioned in a loop between two β-propeller blades [15,16]. The paralogues Nup188/Nup192 have an NTR-like structure [17,18,19] and form the central layer between the membrane binding NUPs and the channel nucleoporin heterotrimer. The rigid layers are linked by flexible connectors: the flexible N-terminus of Nic96 ties the membrane binding layer to Nup188/Nup192 and the channel nucleoporin heterotrimer, and the short linear motifs (SliMs) in the membrane-interacting Nup53/Nup59 connect most of the inner ring NUPs [2,20,21,22,23,24] (Figure 1). While the NUPs within a single inner ring spoke have large interaction surfaces [20], recent structural models of the NPC propose that the interactions between spokes are minimal and instead are mostly mediated by natively disordered and flexible connector NUPs [2,20,21,22]. This flexibility allows the NPC to adjust its diameter depending on the physiological state of the cell [2,5,6,7,25,26], and the resulting spaces between spokes might solve the long-standing question of how transmembrane proteins can pass through the NPC. Intriguingly, Nup188/Nup192 (hsNup188/hsNup205; hs for *Homo sapiens*) not only share structural similarity with NTRs, their interaction with Nic96 (hsNup93) also resembles the interaction of the transport receptor importin-β and the importin-β binding domain (IBB) of its cargo complex [2,5,21,27]. This points to a common evolutionary origin of NUPs and transport receptors [2,17,18,19,28].

### 2.2. Symmetric Outer Rings: The Versatile Outer Coat of the Nuclear Pore Complex

The outer rings on the nuclear and cytoplasmic faces of the NPC are largely identical and made up of rigid subcomplexes known as Y complexes [3,29]. These building blocks are themselves composed of six conserved constituent proteins (Seh1 is not a conserved element of the Y complex in thermophilic fungi and *Aspergillus nidulans* [30,31,32]), which form a structure resembling the shape of the letter Y [33,34,35,36] (Figure 1), and eight Y complexes assembled in a head-to-tail manner. *S. cerevisiae* has a single cytoplasmic and nucleoplasmic Y complex ring [1,2,10], human and *Xenopus laevis* NPCs carry two Y complex rings per side [3,16,27,37,38,39], and the green algae *Chlamydomonas reinhardtii* and fission yeast *Schizosaccharomyces pombe* exhibit an asymmetric distribution, with two nuclear and only one cytoplasmic Y complex rings [4,7]. Notably, the cytoplasmic Y complex ring of *S. pombe* only consists of the Y triskelion, breaking the head-to-tail arrangement observed in other species [7]. Surprisingly, the number of Y complex rings can vary even within the same cell: a subset of NPCs with two nucleoplasmic Y complex rings was recently observed in budding yeast ([2], further discussed below).

The largely α-solenoid core of the Y complex is tethered to the NE by ALPS motifs in the β-propeller of Nup120 and Nup133 [40,41,42] and decorated by several species-specific β-propeller NUPs [43,44,45]. The α-solenoid/β-propeller architecture of the outer and inner ring NUPs is similar to the vesicle-coating protein complexes COPI and COPII, and the β-propeller protein Sec13 is a shared component of both NPCs and COPII complexes, suggesting a common evolutionary origin (reviewed in [46]). Although the eightfold rotational symmetry of the NPC is well established, deviations have been observed in *X. laevis* NPCs [47,48], which raises the question how the eightfold symmetry of the NPC is formed. Since the connections between inner ring spokes are flexible and the inner ring diameter can change drastically [5,6,7,25], it seems likely that the oligomerization of the Y complex ring plays a key role in establishing the correct stoichiometry of NPC subunits. However, the Y complex itself is not a rigid structure and has multiple hinge points [33], and additional constraints, such as, e.g., membrane interaction may thus be needed to determine the eightfold symmetry of the NPC.

The outer rings are connected to the inner ring by a set of paralogues with flexible linkers (Nup116, Nup100, Nup145N) [10,21,23,49], and the double rings in metazoan and *C. reinhardtii* NPCs are linked by an additional copy of hsNup155 or crNup155 (cr: *C. reinhardtii*), respectively [3,4,16,37] (Figure 1). In metazoan NPCs, the chromatin-binding NUP ELYS is associated with one short arm of the Y complex on the nucleoplasmic side [5,27,50,51,52] (Figure 1).

Interestingly, recent biochemical characterization and higher-resolution electron microscopy (EM) maps of the NPC revealed that the importin-β-IBB-like complex hsNup205-hsNup93 is not only a part of the inner ring, but it can also be found in the outer rings of metazoan NPCs [5,22,27,39]. A characteristic question mark-shaped density can also be seen in EM maps from the double Y complex rings of *S. cerevisiae* and *C. reinhardtii* [2,4], and the presence of the hsNup205-hsNup93 heterodimer and its homologues may thus be conserved and important for the oligomerization of double Y rings.

### 2.3. Asymmetric Appendages: Functional Extensions of the Nuclear Pore Complex

The symmetry of the outer rings is broken by several subcomplexes that specifically bind to the cytoplasmic or nucleoplasmic ring. Identified by classical EM experiments, the cytoplasmic filaments and nuclear basket are the most prominent asymmetric components of the NPC [8,9,53]. The term cytoplasmic filaments is often used as a synonym for all NUPs that preferentially localize to the cytoplasmic side of the NPC. However, the main component of these elongated filaments protruding into the cytoplasm in metazoa is the largely disordered C-terminus of hsNup358, which harbors multiple Zinc-fingers and Ran-binding domains, and plays an important role in receptor-mediated transport and protein translation [22]. hsNup358 is specific to metazoa and stabilizes the cytoplasmic double Y ring [22,37].

The majority of the other cytoplasmic NUPs are conserved across species and form the so-called mRNA export platform. This extends to the center of the NPC [1,2,54,55,56] and plays a key role in mRNA export and remodeling [57]. Intriguingly, the mRNA export platform has high similarity with the channel nucleoporin heterotrimer at the center of the NPC, with Nsp1 being a shared component between the two. Further, the positioning of the hsNup93-hsNup205 heterodimer in the cytoplasmic outer ring and its biochemical interactions suggest that hsNup93 connects the cytoplasmic mRNA export platform in a similar way as the channel nucleoporin heterotrimer in the inner ring [22] (Figure 1). Interestingly, the mRNA export platforms in metazoa and yeast have different overall architectures. In yeast, the cytoplasmic coiled-coil NUPs form a single complex, whereas two parallel-orientated complexes are present in the *X. laevis* NPC [27]. This corresponds to the number of cytoplasmic Y rings in the two species. Intriguingly, the mRNA export platform is entirely absent in the more divergent eukaryote *Trypanosoma brucei* [13,58]. In contrast to the conserved Y complex and inner ring, the mRNA export platform might thus have specialized to meet the needs of the respective organism during evolution.

The nuclear basket was identified in early EM studies because of its characteristic elongated structure [8,9], but due to its flexible nature, it remains one of the least structurally characterized modules. The majority of the basket-like structure seen by classical EM analysis [59,60] likely stems from the large coiled-coil hsTPR (*S. cerevisiae* Mlp1/Mlp2) [61,62]. Although the stoichiometry of the nuclear basket coiled-coil NUPs is not entirely clear [10,63,64,65,66], up to eight basket-like filaments protruding into the nucleoplasm and tethering proteasomes to the NPC have been observed at single NPCs of *C. reinhardtii* [67].

So far, the best-resolved fragments of the nuclear basket are coiled-coil segments that likely belong to Mlp1/2 and bind to the nuclear Y complex [2], which is consistent with other EM and crosslinking data [1,10]. The inventory of the *S. cerevisiae* nuclear basket is completed by the mostly disordered Nup1, Nup2, and Nup60. Although these NUPs have evaded structural characterization, biochemical studies show that Nup1 and Nup60 (hsNup153) interact with the NE via an amphipathic helix (AH) [68,69,70]. Similar to the linker NUPs in other subcomplexes, Nup60 flexibly connects the nucleoplasmic Y complex ring with the Mlps and Nup2 (hsNup50) via SLiMs [68,71,72] (Figure 1). Further, Nup1, Nup2, and Nup60 contain FG repeats and, together with the Mlps, are important for export and quality control of mRNA (reviewed in [73]).

### 2.4. The Membrane Ring: An Enigmatic Girdle

Besides the membrane interactions of the inner and outer rings mediated by ALPS motifs, the NPC is also directly anchored in the NE by transmembrane NUPs. Because of their transmembrane regions, it is difficult to purify these proteins or distinguish them from the NE in EM studies, and the structure of the membrane ring is thus poorly characterized. In *S. cerevisiae*, there are four transmembrane NUPs, which are not as highly conserved as other components of the NPC [74] (reviewed in [75]). Only Ndc1 has a well-defined ortholog in metazoa [76,77], and is the only essential protein of this group. Ndc1 interacts with the inner ring NUPs Nup53/59 and Nup170 (in humans: hsNdc1, hsNup35, hsNup155, and additionally ALADIN) to form a membrane interaction hub that anchors the inner ring to the NE [2,5,24] (Figure 1). Pom152 and the human Gp210 are the only NUPs with structured domains in the NE lumen: both contain a series of luminal immunoglobulin repeats [78,79,80]. Despite the low primary sequence conservation and different membrane topology, the high structural similarity could hint at a common origin for both proteins. The immunoglobulin folds of Pom152 form a belt-like chain of beads around the NPC in the NE-lumen, which is anchored near the membrane interaction hub [2,7,10,79,80,81]. The belt-like luminal ring deforms together with changes in NPC diameter [2,7], which raises the possibility that it regulates the diameter of the NPC. However, neither Pom152 nor Gp210 is essential [82,83], and deletion of Gp210 does not lead to variation of the NPC diameter in cellulo [5]. Furthermore, the expression level of Gp210 in different cell lines varies widely [65,84], suggesting a more intricate role of the luminal ring than as just a mechanical girdle.

### 2.5. Linker Nucleoporins: An Invisible Thread Stitching the Nuclear Pore Complex Together

The NPC embodies two seemingly contradictory properties. On the one hand, it uses rigid building blocks with large interaction surfaces to form stable subcomplexes, such as the Y complex and the inner ring spokes, which confer a high degree of stability to the NPC core in post-mitotic cells [85,86,87,88,89] (reviewed in [90]). On the other hand, its structural flexibility allows for drastic changes in diameter [2,6,7,25] and likely enables a fast assembly and disassembly of the NPC in open mitosis [91]. How can these properties coexist in one structure? The emerging solution is a peculiar mode of association between the different NPC modules via intrinsically disordered NUPs. Homologues of the *S. cerevisiae* FG repeat NUPs Nup100, Nup145N and Nup116, and non-FG NUPs Nup53 and Nup59 are universally capable of linking several NPC elements each via SLiMs spread throughout their intrinsically disordered domains [14,23,92,93]. In this way, each of them can flexibly join several core subunits, akin to beads on a string (Figure 1). The electron densities observed next to the core NUPs in high-resolution NPC maps and chemical crosslinking data all point to SLiM-mediated connectivity of the NPC subunits [2,5,10,21,22]. Further, flexible connections could arise from the ability of some core NUPs to directly bind FG repeats [49]. These multivalent interactions might create a velcro-like effect that brings about both stability and structural plasticity (Figure 1).

The interactions via short motifs are a prevailing theme also outside the NPC core. Short motif interactions contribute to the attachment of the mRNA export platform, central channel NUPs, the nuclear basket and transmembrane NUPs [22,23,24,68,69] (Figure 1). Interestingly, the interactions of the NPC with the NE rely on the same principle. Although transmembrane NUPs are one of the least evolutionarily conserved groups [74] (reviewed in [75]), short lipid-binding AHs found within multiple core and non-core NUPs are a conserved feature, often seen positioned along the lipid membrane in current NPC models [2,5]. This multitude of binding sites could stabilize the high curvature induced in the lipid membrane and establish a tight association of the NPC spokes to the pore membrane (reviewed in [94]).

Taken together, the short interaction motifs and intrinsically disordered domains emerge as key elements of NPC connectivity.

## 3. Nuclear Pore Complex Assembly: Many Roads to the Same Destination

Growing and proliferating cells must produce new NPCs to cope with increasing demands in nucleocytoplasmic communication. Non-dividing cells also assemble new pores in order to replace old ones [88]. But how is NPC assembly orchestrated and which factors control it in space and time? To create a new NPC, individual NUPs must fold, find their correct interaction partner(s), and become integrated into the double lipid membrane as an oligomeric multiprotein assembly. These events must be coordinated to avoid the formation of faulty structures. Surprisingly—in spite of its high architectural complexity—there are different pathways directing NPC assembly. In metazoa with an open mitosis, a concerted wave of NPC assembly occurs in a timeframe of only a few minutes during mitotic exit, concomitant with reformation of the sealed NE [91,95,96,97,98]. In contrast, NPC assembly during interphase requires perforation of the intact nuclear membrane and is kinetically slower [99,100,101,102,103,104]. Other routes to NPC formation have been reported in specific developmental stages in multicellular organisms. In *Drosophila melanogaster* embryos, NPC-like structures are embedded into cytoplasmic membrane cisternae (*annulate lamellae*), which can fuse with the NE to supply new NPCs [105], while NPC assembly in *D. melanogaster* oocytes involves large liquid-like condensates of stockpiled NUPs [106].

Due to the synchrony of a large number of assembly events, NPC assembly at the end of mitosis has proven particularly amenable to experimental investigation (recently reviewed in [107]). Early during mitotic exit, NPC assembly initiates with chromatin-bound NUP assemblies which are integrated into membrane fenestrae of the reforming NE [103,108]. In contrast, NPC assembly into a sealed NE during late mitosis and in interphase occurs via an “inside-out” mechanism that initiates with the deformation of the inner nuclear membrane and ultimately requires fusion of the inner and outer nuclear membranes [109] (reviewed in [110]) (Figure 2). Besides being significantly slower (hour versus minutes timescale), interphase assembly also differs in the order of NUP recruitment [100,101,102,104]. In addition, interphase and mitotic assembly modes diverge significantly in their functional requirements. Assembly during interphase in vertebrates specifically depends on the nuclear basket NUP hsNup153 and the transmembrane NUP hsPom121, while the chromatin binding NUP ELYS and the reticulon-like protein REEP4 are important for assembly at the end of mitosis [70,111,112]. Moreover, the recruitment order of NUP subcomplexes during inside-out assembly may differ between lower and higher eukaryotes: as judged by metabolic labeling analysis, in yeast, it begins with the symmetrical core NUPs and ends with the late recruitment of the nuclear basket NUPs Mlp1/2 [102]. Conversely, inside-out assembly during late mitosis in mammalian cells is characterized by late recruitment of the central channel NUPs [104].

The versatility of NPC assembly may be rooted in the modular principle of its organization, which provides the opportunity for variations of a core assembly mechanism through the bypassing of individual steps. For example, NPC assembly during mitotic exit can proceed faster because it does not require membrane fusion and can rely on a large pool of preassembled NPC subcomplexes [96,103]. In addition, the Y complex and some inner ring NUPs were recently reported to remain associated with each other and with membranes throughout open mitosis, which would further expedite assembly [96]. Furthermore, differences in assembly order and mechanistic requirements may be governed by the relation of NUPs to chromatin and membranes. While membrane and nuclear subunits take center stage in inside-out assembly [70,111,113,114], the Y complex, with its interactions to both chromatin and membranes, is a key player in the reformation of NPCs and nuclear membranes in open mitosis [52,112,115].

The location of the NPC at the border between the nuclear and cytoplasmic compartments poses significant logistical challenges to its assembly. It is puzzling how—both in interphase and at the end of mitosis—NPC assembly favors nuclear membrane over cytosolic ER membranes. Moreover, NPC assembly into a sealed membrane requires fusion of the inner and outer nuclear membranes. This fusion has to take place within the NE lumen, and surprisingly, the cellular machinery capable of that has not yet been identified. The fusion event also has to be coordinated with establishment of the NPC permeability barrier to avoid compromising the compartmentalization of the nucleus.

### 3.1. From Nascent Polypeptides to Nucleoporin Subcomplexes

The NPC is organized into distinct subcomplexes (see “Section 2”). Analysis in budding yeast indicates that newly translated NUPs initially co-assemble within the subcomplexes [102]. Unlike for other well-studied structures of comparable size and complexity, such as the ribosome, the proteasome or the mitochondrial respiratory chain, there is no firm evidence that folding or assembly of NUPs into subcomplexes requires specific assembly factors.

Evaluation of NPC assembly kinetics in budding yeast suggests that newly made NUPs typically associate with their immediate interaction partners within minutes [102], which is comparable to the timescale of protein translation [116]. One mechanism that could account for the fast assembly kinetics of some NPC modules and the lack of dedicated assembly factors is cotranslational assembly. A classical example is the autoproteolysis-mediated generation of the Y complex NUP Nup145C and linker Nup145N (hsNup96 and hsNup98 in vertebrates) that form a non-covalent complex [117,118]. Cotranslational interactions were also recently reported for other linker NUPs and for several constituents of the well-structured channel nucleoporin heterotrimer and Y complexes [119,120].

Analogous to other macromolecular complexes, such cotranslational interactions may assist in the folding and correct association of interacting NUPs as soon as the nascent polypeptides emerge from the ribosome [121,122,123] (reviewed in [124]). Specifically for the NPC, it has been suggested that cotranslational interactions prevent the erroneous assembly of paralogous NUPs that share similar interaction properties [120]. Such a mechanism could have been adopted during NPC evolution, when NUP diversity arose through multiple gene duplication events from a few ancestral genes [125,126] (reviewed in [46]).

The lack of sophisticated machinery to aid NPC assembly is surprising but somewhat consistent with the view that the NPC shares common evolutionary roots with COP coats [125,126] (reviewed in [46]). It is conceivable that the NPC might share self-assembly characteristics inherent to COP coats, where large COP structures are produced by the repetitive addition of simple coatomer elements. However, this is a limited analogy that does not account for evolutionary innovations such as the FG repeats or the asymmetrical NPC modules for which specific assembly factors might thus far have evaded identification.

Specific chaperones might nevertheless contribute to NPC assembly events. Cells depleted of torsins, which are members of the luminal AAA+ ATPase superfamily proteins, develop NE herniations that likely represent stalled NPC assembly intermediates. Such misassembled NPCs accumulate a set of factors including myeloid leukemia factor 2 (MLF2) and chaperones of the Hsp70 and Hsp40 families DNAJB2, DNAJB6, HSC70, and HSPA1A [127,128,129]. Conversely, DNAJB6-depleted cells display NPC-like structures in cytoplasmic *annulate lamellae* [128]. The role of these factors in NPC assembly is not clear, but NUP FG repeats are one of the likely targets, since some of them accumulate in the herniations in an FG NUP-dependent manner and can bind FG repeats [128,129]. Moreover, DNAJB6 can prevent aggregation of the FG repeats in vitro [128,129]. It is possible that this chaperone activity contributes to the dynamic interactions of FG repeats with core NUPs during NPC insertion, or that it controls the quality of the nucleocytoplasmic diffusion barrier brought about by the FG repeats [128].

### 3.2. Targeting of Nuclear Pore Complex Assembly to the Nuclear Envelope

Although NPCs are normally located exclusively in the NE, excess NUPs can in principle also form NPC-like structures in cytosolic ER membrane sheets, e.g., in *annulate lamellae*, as happens in oocytes and early embryonic cells containing large NUP stockpiles (reviewed in [130]). How is NPC assembly targeted specifically to the NE, which is a direct extension of the cytoplasmic ER network? Multiple in vivo and in vitro studies link NPC biogenesis with the function of the nucleocytoplasmic transport machinery (NTM)—the NTRs and the Ran GTPase system—that direct nucleocytoplasmic exchange. Both in higher and lower eukaryotes, genetic interference with the NTM disrupts NPC assembly [131,132,133]. In vitro, RanGTP and soluble NTRs exert antagonistic effects on NPC assembly both in the NE and in cytoplasmic *annulate lamellae*, with RanGTP suppressing the inhibitory effect of NTRs [70,100,134,135,136,137,138]. Contribution of the NTM is most strikingly illustrated by the ability of bead-immobilized RCC1 (the chromatin-associated guanidine nucleotide exchange factor for Ran that generates its active GTP-bound form in the nucleus) to convert the bead volume into a pseudo-nuclear compartment covered by a sealed double membrane containing transport-competent NPCs [139].

Concentration of RanGTP in the vicinity of chromatin provides a spatial cue for nuclear transport and mitotic spindle assembly by governing the assembly of NTR complexes and, through this, the functional properties of nucleocytoplasmic transport cargos and spindle assembly factors (see, e.g., [140,141] for a detailed discussion). A large body of evidence suggests that a Ran-mediated mechanism spatially guides various steps in NPC assembly in a similar way, thus confining the process to the nuclear membrane: in open mitosis, the NTM targets essential NPC modules to the chromatin surface, consistent with NUP assemblies observed on chromatin before membrane enclosure [103,108,142]. This targeting is achieved at least in part through chromatin recruitment of ELYS and, consequently, the Y complex, which depends on ELYS being released from the NTRs importin-β and transportin-1 by RanGTP [111,115,136,143,144,145,146,147]. Moreover, RanGTP promotes—in an NTR-mediated manner—fusion of membranes around chromatin [135,148] and could encompass additional levels of regulation, as illustrated by the important role that the stimulation of hsRCC1 by the basket NUP hsNup50 plays during mitotic NPC assembly [149]. The role of the NTM during interphase NPC assembly is less clear. One attractive hypothesis is that it promotes the import of NUPs through existing NPCs (Figure 2). This nuclear sequestration would hard-wire NUP targeting to the nuclear membrane into the inside-out assembly pathway. Indeed, two NUPs specifically required for interphase NPC assembly in vertebrates, the transmembrane NUP hsPom121 and the nuclear basket NUP hsNup153, rely on NTM-mediated import to reach the nucleus [70,111,113,150]. In the case of hsNup153, its NPC assembly function specifically requires nuclear import-coupled membrane binding through release from the NTR transportin-1 [70]. A similar NTM-mediated mechanism might target membrane protein Pom33 to the yeast NPC [151].

However, there is no evidence that large symmetrical core NUPs contain functional nuclear localization sequence motifs, and the majority of them exceeds the NPC diffusion limit, with molecular weights higher than 100 kDa for single NUPs and up to 1 MDa for assembled modules such as the Y complex. How could the logistical challenge of their nuclear delivery be overcome? Interestingly, not only do many NUPs show structural similarities with NTRs (see “Section 2.1”), but many symmetrical core NUPs can also directly bind to FG repeats and can pass through the intact NPC by facilitated diffusion akin to bona fide NTRs [17,49,152]. It is therefore possible that the core NUP modules are delivered to the NPC assembly sites within the nucleus analogous to some transmembrane proteins—by a diffusion-retention mechanism dictated by the availability of binding sites inside the nucleus [153,154]. It will be interesting to investigate whether other NUP classes, such as FG NUPs, can also pass through the intact NPC.

The NTM also contributes to NPC assembly in the nucleus by regulating binding between NUPs. For example, the release of Kap121 from Nup53 in the nucleus by the activity of RanGTP frees the binding site of Nup170 [155], and Kap60 modulates the interactions between the nuclear basket NUPs Nup60 and Nup2 in a RanGTP-dependent manner [156]. Interestingly, Nup60 and Nup2 can also bypass the need of NTRs and directly bind RanGTP, which enhances their association [156]. These NTM-controlled NUP binding events could trigger further NUP association steps in the nucleus, e.g., similar to interaction between hsNup155 and hsNup93, which is promoted by the membrane association of hsNup155 [157].

Spatial control of NPC assembly might also be contributed by other mechanisms such as post-translational modifications (PTMs). It is well established that NPC connectivity can be disrupted in open mitosis by NUP phosphorylation through mitotic kinases such as Cdk1, NIMA-related kinases or PLK1 [2,114,158,159,160,161]. This mechanism might not only time mitotic NPC dis- and re-assembly, but also act as a spatial cue in concert with the NTM. Supporting this view, ELYS contains a docking site for the major protein phosphatase PP1 that is required for its chromatin targeting and proper NE assembly [162]. Likewise, hsNup153 is a target of PP1 [163] and mediates post-mitotic chromatin targeting of the PP1 adaptor Repo-Man needed for timely chromatin decondensation [164,165]. It remains to be understood to what extent mechanisms such as localized dephosphorylation activity at chromatin could also contribute to spatial control of NPC assembly.

### 3.3. Creating Functional Nucleocytoplasmic Conduits

A central challenge in NPC assembly is the perforation of the NE to form a nucleocytoplasmic channel. This requires fusion of the inner and outer nuclear membranes. Native assembly intermediates observed by EM evidence that this is initiated by the formation of a shallow dimple in the inner nuclear membrane that consists of octagonal rings resembling the NPC symmetric core modules [109,166]. That the membrane fusion event is likely preceded by the assembly of the symmetrical core is also suggested by early recruitment of symmetrical core NUPs during native assembly in budding yeast [102] and by the analysis of various stalled NPC assembly phenotypes both in higher and lower eukaryotes [1,49,127,128,129,166,167,168,169,170,171,172,173]. Interestingly, stalled NPC assembly is often associated with nuclear membrane herniations—structures morphologically resembling NPCs and sealed by the nuclear membrane (reviewed in [110]) (Figure 2). Indeed, structural characterization of herniations in Nup116-deficient yeast cells revealed presence of all major NPC features except for the cytoplasmic outer ring and the mRNA export platform [1]. It is tempting to speculate that assembly of the NPC core confers a checkpoint that ensures an intact diffusion barrier prior to perforation of the NE. The accumulation of electron dense material, and K48-ubiquitylated proteins observed in herniations [127,173,174] might point to transport-competence of NPC assembly intermediates.

The formation of NPC-like precursors requires strong deformation of the inner nuclear membrane. The precursor must create both convex and concave curvatures (in the nuclear membrane plane and along the nucleocytoplasmic axis, respectively). It also has to generate a concave dome-shaped dimple in the inner membrane. It is not fully understood what forces produce such complex membrane deformations. Lipid-binding AHs are common protein motifs that both generate and sense membrane curvature (reviewed in [175]). Such motifs found within multiple NUPs are an emerging theme in NPC biogenesis. Both in higher and lower eukaryotes, AHs of core and linker NUPs important for NPC biogenesis have been shown to sense concave membrane curvature (reviewed in [94]) and could potentially shape the concave membrane around the NPC assembly site. In addition, reticulons and reticulon-like proteins, which are wedge-shaped membrane-curving proteins, could contribute as well [112,176,177]. Different mechanisms may generate convex curvature. First, this could be achieved through asymmetric distribution of lipids in the lipid bilayer (reviewed in [178]). In yeast, phosphatidic acid (PA) accumulates at stalled NPC assembly sites [179] and accumulation of PA at the NE can be readily induced by overexpressing the NE/ER transmembrane protein Apq12 implicated in NPC membrane fusion [180]. These lipids with small headgroups can promote curvature by unequal distribution between the two layers of the lipid membrane (reviewed in [181]). Second, liquid-liquid phase separation of intrinsically disordered domains could act as a driver of membrane deformation [182,183]. It is intriguing to speculate that liquid-liquid phase separation of cohesive FG repeats or other natively disordered NUP domains could act in concert with altered lipid composition and membrane-binding motifs to shape the membrane and prime it for fusion.

The mechanism of pore membrane fusion remains elusive. Membrane fusion cannot occur spontaneously, requiring fusogenic proteins to overcome associated energy barriers (reviewed in [184]). In yeast, three structurally related and interacting transmembrane proteins, Brl1, Brr6, and Apq12 recently came into the spotlight. None of them constitutively associates with NPCs, but their depletion stalls NPC assembly, producing characteristic NE herniations [168,169,170,171,172,180,185]. How Brl1 and its interaction partners promote pore membrane fusion is not clear. At least two of them, Apq12 and Brl1, depend on luminal lipid-binding AHs for their functionality [169,172,180]. Speculatively, the AHs could directly tether opposing membrane leaflets similar to some viral fusogens (reviewed in [186]) and/or distort lipid packing similar to membrane-lytic peptides (reviewed in [187]) (Figure 2). Alternatively, they could facilitate fusion by locally altering lipid membrane composition. Indeed, the functionality of the Brl1/Apq12/Brr6 triad is strongly influenced by altered lipid metabolism and biophysical properties of membranes [168,170,171], and Apq12 can induce NE enrichment of PA lipids [180] implicated in various membrane fusion processes (reviewed in [181]).

By mediating membrane fusion, the Brl1/Brr6/Apq12 triad might play the role of “assembly sensors” that couple membrane piercing with maturity of NPC precursors to guarantee seamless NPC insertion. Supporting this view, Brl1 overexpression rescues NPC biogenesis in GLFG repeat deficient assembly mutants [49,188]. Further, Brl1 can suppress nuclear export machinery defects, suggesting a deeper connection between pore membrane fusion and the nucleocytoplasmic transport [189]. It will therefore be important to systematically analyze the functional relationships of Brl1/Brr6/Apq12 with NUPs, lipid composition, and the nucleocytoplasmic transport machinery.

Although the Brl1/Apq12/Brr6 triad is essential in yeast, no homologues are found in higher eukaryotes. Instead, in higher eukaryotes, similar NPC assembly defects were linked to torsins, the nuclear membrane-associated AAA+ ATPases that are in turn absent from yeast [127,190,191,192]. The mechanistic role of torsins is not known. The luminal localization of ATPase domains and the central role of the AAA+ ATPase NSF in SNARE-mediated fusion of cytosolic membranes [193] make them attractive candidates for the vertebrate NPC fusogenic machinery. Since the defective NPCs in torsin-deficient cells accumulate a subset of proteins, including ATP-dependent chaperones (see “Section 3.1”), they could potentially contribute to pore formation as well. In sum, it appears that the formation of the nucleocytoplasmic conduit can be executed differently in different species.

The herniation phenotype characteristic for stalled NPC assembly is also observed in mutants with defective components of the ESCRT-III machinery [194,195]. Although these factors have been primarily attributed to NPC surveillance, their contribution to pore membrane fusion cannot be ruled out.

### 3.4. Maturity: Compositional and Functional Variation of the Nuclear Pore Complex

The fully assembled NPC is a very stable structure. Components of the inner and outer ring complexes are not exchanged within one cell cycle [86,196] and can last weeks or months in non-dividing cells [85,87,88,89]. In contrast, peripheral components, e.g., of the nuclear basket are more dynamic and exchange readily with a soluble pool on a timescale of minutes [86,196]. Even more dynamic are the transport factors and possibly additional effector proteins involved in the many functions of the NPC as an organizing hub at the NE [102,196,197]. The NPC thus combines a stable scaffold with dynamic effector proteins.

The modularity of the stable NPC core with its flexible connectors and highly redundant interactions likely provides the framework that supports the surprising diversity observed in NPC structure and composition across species (recently reviewed in [198] and compare above). Moreover, recent work has started to elucidate the extensive compositional variability of NPCs within individual species and even within individual cells, as well as the pathways that regulate their function. To date, the major source of variability described in NPCs is in the make-up of peripheral NUPs. For example, the hsTPR-homologous nuclear basket proteins Mlp1/2 are not present in all NPCs in budding and fission yeast, and NPCs that do contain them are excluded from certain regions of the NE [2,199,200,201,202,203]. Furthermore, aged budding yeast cells accumulate clusters of NPCs which lack several nucleoplasmic and cytoplasmic NUPs [204,205]. Intriguingly, recent work indicates that variation is not restricted to peripheral components, since budding yeast NPCs can contain either one or two nucleoplasmic Y complex rings [2]. Importantly, the outer rings act as attachment sites for most peripheral NUPs, and differences in outer ring organization may thus directly influence and regulate the association of peripheral NUPs and their interactors [2,206].

NPC isoforms can also represent age-specific subpopulations, as exemplified by budding yeast, where a significant fraction of NPCs does not contain the basket NUPs Mlp1 and Mlp2 [200,203]. These NUPs were recently shown to associate with NPCs only very late during the NPC maturation process [102], suggesting that the NPC subpopulation lacking Mlp1/2 constitutes recently assembled NPCs. A similar kinetic mechanism could also regulate the fraction of NPCs that assemble a second nuclear Y ring [2]. Intriguingly, loss of the Mlp1/2 homologue hsTPR in 50% of NPCs was observed in mammalian cells upon depletion of hsNup133 [206], suggesting that kinetics might also regulate basket assembly in higher eukaryotes.

A possible mechanism leading to compositional differences between cell types is modulation of the expression levels of NUPs. Peripheral and membrane NUPs in particular exhibit significant variability in expression levels across different cell types [65,207,208,209]. However, more acute modifications of NPCs, e.g., during stress response signaling [210,211,212,213], differentiation [211] or in relation to the cell cycle [202,214,215], require regulatory mechanisms that can act on shorter time scales and are potentially restricted to subsets of NPCs. Two such mechanisms have been described in the generation of NPC variants: PTMs and proteolytic cleavage.

Phosphorylation has long been known to regulate the disassembly of NPCs during open mitosis [114,159,160,161], as well as partial NPC disassembly during semi-open mitosis in the filamentous fungus *A. nidulans* [158]. Certain stress conditions also result in the phosphorylation, ubiquitination, and SUMOylation of NUPs, in particular those in the nuclear basket [216,217,218], and these modifications can regulate the interaction between NUPs [217,219]. Furthermore, acetylation of the nuclear basket NUP Nup60 was recently implicated in the generation of modified NPCs in budding yeast [205,214,215]. These PTMs are likely only a small fraction of regulatory modifications involved in regulating NPC function, and more work is needed to identify and characterize PTMs on NUPs.

Acute changes to NPC composition can also be induced by proteolytic cleavage. In the early stages of apoptosis, several NUPs are targeted by caspases [211,220,221,222,223,224], leading to the removal of the cytoplasmic filaments and the nuclear basket [225] and to the loss of NPC barrier function [226]. Intriguingly, caspase-dependent degradation of a set of peripheral NUPs was recently reported to also occur during cellular differentiation and ER stress [211].

The functional consequences of variation in NPC composition are still largely unknown, but they may underlie tissue-specific effects observed in diseases associated with mutations in NUP genes (reviewed in [227]) and cell type-specific susceptibility to infection by pathogens such as HIV-1 [209]. In general, three categories of functional effects can be observed. First, modulation of the NTR complement at the NPC can regulate the available transport pathways. Such effects have been reported for mRNA and protein export [211,212,213,214,215]. For example, the budding yeast mRNA export factor Sac3 is released from the nuclear basket in newly budded daughter cells, which leads to an inhibition of mRNA export [214], and similarly, bulk mRNA export is inhibited by the release of the cytoplasmic mRNA export factor Dbp5 from the NPC during glucose starvation [212]. Interestingly, the availability of transport cargo may also influence NPC composition, as interference with mRNA transcription or 3′ end processing perturbs the association of the budding yeast nuclear basket NUP Mlp1 with the NPC [200].

Second, variant NPCs can exhibit differences in their function as scaffolds that link to chromatin, the cytoskeleton or signaling effectors. For example, in yeast cells, acetylation modulates the interaction of NPCs with chromatin loci and extrachromosomal circles [204,215,228,229], and ubiquitination controls the interaction with the dynein light chain Dyn2 [219]. Intriguingly, some recent studies that link components of the inner ring to silenced chromatin in yeast and *D. melanogaster* suggest that these interactions might not always occur in the context of a channel-forming NPC but might in some cases involve alternative NUP complexes in the NE [230,231,232].

Third, the permeability of the NPC can be affected. This occurs in certain species such as *A. nidulans* during semi-open mitosis [158,233] or in transient stages of *S. pombe* meiosis [234,235], but also during apoptosis [226] or ageing [85]. In light of this, it is conceivable that there might be additional conditions where NPC permeability and thus compartmentalization of the nucleus could be transiently compromised.

In the past years, it has thus been clearly established that not all NPCs are equal. Future work will undoubtedly uncover additional variants and their functional specialization as transport channels and interaction platforms.

## 4. Nuclear Pore Complex Remodeling and Functional Maintenance

The exceptional stability of the NPC core in post-mitotic cells [85,87,88,89] raises the question of how the functionality of the complex is maintained, and which mechanisms allow detection of malfunction. Which pathways contribute to NPC repair and how is disturbed NPC function associated with disease?

### 4.1. Rejuvenation

Dividing mammalian cells naturally renew their NPCs by re-assembling them after each cell division. Interestingly, dedicated renewal mechanisms during cell division also exist in cells with closed mitosis, where NPCs remain intact. As with other damaged components (reviewed in [236]), *S. cerevisiae* has evolved mechanisms to retain potentially damaged NPCs in the mother cell: while approximately 50% of assembled NPCs are passed on to daughter cells during normal mitosis [237], different classes of defective NPCs are retained in the mother cell by a barrier at the bud neck [195,205,238,239], which can only be surpassed by an active mechanism that depends on the essential FG NUP Nsp1 [238,240]. This quality control step contributes to the birth of a rejuvenated daughter cell.

NPC renewal may also be essential to the meiotic rejuvenation of budding yeast cells. During gametogenesis, pre-existing NPCs are sequestered in an NE compartment that is separated from newly forming spore nuclei and degraded by autophagy during late stages of spore formation [241,242]. Interestingly, the only NUPs that escape this destruction are the dynamically exchanging NUPs Nup1, Nup2, and Nup60 [241].

It is unknown whether NPCs are renewed also in other meiotic or mitotic models with different modes of closed and semi-open nuclear division. For example, symmetric closed mitosis in the fission yeast *S. pombe* involves the removal of a subset of NPCs during NE abscission [201,202], and it will be interesting to test whether this process also contributes to the clearance of defective NPCs.

### 4.2. Repair

Post-mitotic and quiescent cells require different mechanisms to maintain functional NPCs. Many peripheral components of the nuclear pore complex, including the transport receptors, several nuclear basket components, and the transmembrane NUP Ndc1, are rapidly exchanged with a soluble pool [86,102,196] which provides an opportunity to replace damaged subunits with newly synthesized ones (Figure 2). A similar mechanism of renewal might also exist for stable components of the NPC core. For instance, experiments monitoring the exchange of subunits in quiescent mammalian cells detected chimeric NPCs that contain both old and new copies of hsNup93 [88]. This exchange of subunits could occur after spontaneous dissociation of individual proteins and subcomplexes, but may also follow the ubiquitination and proteolysis of faulty components. Indeed, proteasome-mediated degradation of individual protein subunits within the complex can be experimentally induced while leaving the overall structure of the NPC intact, even in the case of stable core NUPs [6,22,196,243,244]. The fast kinetics of degradation observed in these experiments further lends credence to the idea that ubiquitination and proteolysis can occur directly at the NPC [6,22,196,243,244]. Although experimentally induced degradation of a few core NUPs (notably hsNup96 and hsNup93) leads to significantly compromised NPC structures [6,243], the high redundancy of connections in the NPC still supports the removal of individual copies of these NUPs in the context of the intact pore. Interestingly, the interaction of proteasomes with NPCs and in particular with nuclear basket components has been described in different model systems [62,67,245], and it is tempting to interpret these NPC-associated proteasomes as dedicated guardians of protein quality, not only for NPC cargo but also the NPC itself.

How could defective NUPs be recognized and marked for proteasomal degradation? A surveillance pathway for NPCs that involves the ESCRT-III machinery has been characterized in *S. cerevisiae* [194,195,246,247,248,249]. Mutants in the ESCRT-III ATPase Vps4 accumulate abnormal NPCs [195,246] and are defective in proteasome-mediated degradation of NUPs in an NPC assembly mutant background [195]. ESCRT-III may thus be involved in the recognition of faulty NPCs or nucleoporins and signal their removal via the proteasome.

An alternative pathway for the ubiquitination of membrane NUPs in budding yeast is “inner nuclear membrane associated degradation” (INMAD). This pathway relies on the Asi1-3 complex, a dedicated transmembrane E3 ubiquitin ligase at the inner nuclear membrane [250,251], and was recently shown to target the NPC-associated paralogous proteins Pom33 and Per33 for degradation [252]. What role this pathway plays in NPC surveillance, and which degradation pathway monitors inner nuclear membrane protein homeostasis in higher eukaryotes, remains to be discovered.

However, rather than being targeted for degradation, misfolded protein domains can also be substrates to chaperones that can help them refold. Several lines of evidence point to the role of classical chaperones in the maintenance of functional NPCs in yeast, although most of this evidence is circumstantial. For example, overexpression of Ssa1, a cytosolic Hsp70-type chaperone, can suppress certain mutations that lead to nucleocytoplasmic transport defects in *S. cerevisiae* [253], and the ER-associated Hsp70 co-chaperone Snl1 is functionally linked to NPC biogenesis defects caused by deletion of the FG NUP Nup116 [254]. More recently, the soluble Hsp70 co-chaperones DNAJB6 and DNAJB2 were implicated in interphase NPC assembly in vertebrate cells [128]. The exact role of these chaperones in NPC maintenance remains unclear, but their targets might be the intrinsically disordered FG repeats, which tend to rapidly collapse into non-physiological solid aggregates in vitro [255,256]. Indeed, DNAJB6 and DNAJB2 display disaggregation activity towards FG repeats of NUPs in vitro [128] and might thus act as sensors for and keepers of the state of FG repeats in vivo. A glycosylation present on many NUPs across metazoa, O-linked N-acetylglucosamine (O-GlcNAc), may also contribute to functional FG repeat domains and NUP stability, since a reduction of O-GlcNAc modifications was observed to promote proteasome-mediated turn-over of these NUPs [257,258,259]. The large number of intrinsically disordered domains may pose a particular challenge to NPC homeostasis and multiple pathways may contribute to their maintenance.

### 4.3. Degradation

While dynamic exchange, targeted degradation, and refolding can solve the problem of damage to individual NUPs, circumstantial evidence for the removal of entire NPCs from the intact NE during interphase stems from observations in tissue culture cells [88,101]. Such events could be mediated by autophagy. Genetic evidence from yeast links multiple components of the autophagy pathway and in particular the ESCRT-III complex to NUPs [195,260]. In budding yeast, two selective autophagy pathways can degrade NUPs during nitrogen starvation and following inhibition of the Target of Rapamycin Complex 1 (TORC1), conditions under which autophagy is upregulated [261,262,263]. Selective autophagy pathways rely on the recruitment of Atg8-containing autophagic membranes by specific autophagy receptors via an Atg8 interaction motif. Autophagy targeting the NE via the specific autophagy receptor Atg39 can contribute to the degradation of NUPs [262,263,264]. In addition, the cytoplasmic NUP Nup159 contains an Atg8 interaction motif [262,263], and can mediate the formation of autophagosomes, which deliver NE-derived vesicles including NPCs to the vacuole [1,261,263] (Figure 2). Due to its cytoplasmic localization, Nup159 is ideally positioned for access by the cytoplasmic autophagy machinery. However, Nup159 is not found in stalled NPC assembly intermediates (NE herniations [1,169], see above) and, indeed, autophagy of NUPs is greatly inhibited in an NPC assembly mutant that accumulates herniations [1,261]. Whether different pathways can degrade NE hernations remains unknown. Interestingly, Nup159 exhibits a tendency to form cytoplasmic punctae, which is exacerbated in cells defective for NPC assembly [24,49,166,168,169,171], and the Atg8 interaction motif in Nup159 can also mediate autophagy of these cytoplasmic clusters [263], contributing to the removal of potentially detrimental cytoplasmic FG repeat-containing aggregates.

It is interesting to speculate how NPC-phagy might contribute to NPC maintenance under normal growth conditions, when TORC1 is active. Under these conditions, NPC-phagy may occur at very low levels and therefore be difficult to detect experimentally. However, the presence of the Atg8 interaction motif on Nup159 raises the possibility that this NUP could signal degradation of individual non-functional NPCs. Further studies will be required to determine conditions under which NPC-phagy occurs, whether it is associated with the specific recognition of damaged components, and whether a similar pathway also exists in higher eukaryotes.

### 4.4. Disturbed Homeostasis and Disease

Although multiple pathways can thus contribute to the repair and removal of defective NPCs, NPC function can become compromised in various diseases and in ageing (reviewed in [265,266]). For example, old cells exhibit changes in the stoichiometry of NPC components in yeast [267,268] and mammals [85,87,89,269] as well as defects in the NPC permeability barrier [85]. Furthermore, several age-associated neurodegenerative diseases are accompanied by defects in nucleocytoplasmic transport and NPC integrity [266,270,271,272,273,274,275]. It remains unclear whether impaired NPC maintenance and function are underlying causes of age-associated diseases and cellular malfunction or rather a downstream consequence of loss of protein homeostasis.

What are the sources of NPC defects, in particular in ageing and neurodegenerative diseases? During ageing, NUPs can gradually lose functionality due to damage accumulated over their long life-time (Figure 2). Indeed, enhanced marks of oxidative damage were found on NUPs in brains from old mice, which correlated with a loss of NPC functionality [85]. However, this is likely not the only source of NPC deterioration, since, for example, in aged budding yeast cells, decline in NPC homeostasis is not accompanied by NUP oxidation [268]. NPC damage might also be caused by irreversible aggregation. Natively disordered NUP FG repeats critical for nucleocytoplasmic transport can undergo irreversible transitions to solid- and amyloid-like states [255,256,276,277]. Since the activity of chaperones might be directly involved in disaggregation of FG repeats [128], such liquid-to-solid transitions could be aggravated in aging cells experiencing a decline in proteostasis [278]. The dynamic state of FG repeats can further be modulated by NTRs [276,279], which can intriguingly also affect the aggregation of neurodegeneration factors such as fused in sarcoma (FUS) [280,281,282,283]. Moreover, NUPs and NTRs co-aggregate with several neurodegeneration-related proteins, such as huntingtin, TDP43 or tau in the cytoplasm, which is accompanied by compromised NPC function [271,272,273,274,284] (Figure 2). NTR homeostasis may thus link NPC malfunction and a variety of neurodegenerative disorders.

Besides general deterioration of the NPC, specific mechanisms of NPC homeostasis can also go awry. In amyotrophic lateral sclerosis (ALS) and frontotemporal dementia (FTD), NPC decay appears to be initiated by the loss of the membrane NUP hsPom121 [285]. Intriguingly, loss of hsPom121 and other NUPs coincides with nuclear localization of the ESCRT-II/III factor CHMP7 [286]—reminiscent of the observation that NPC renewal in quiescent cells relies on both hsPom121 and ESCRT-III machinery [88]. The ESCRT-III machinery and specifically Chm7 were also implicated in NPC quality control in yeast cells [194,195,246,247,248]. It will thus be of high interest to further explore the contribution of NPC quality control to pathogenesis of neurodegenerative disorders.

## 5. Concluding Remarks

Technical advances have recently brought major breakthroughs in our understanding of NPC architecture. Yet many questions about its life cycle, evolutionary origin, and function remain to be answered. Recent structural insight has revealed evolutionary connections of NUPs with the membrane trafficking and nucleocytoplasmic transport machinery. What is the nature of the common ancestral proteins and how did they give rise to modern NPCs? How can the NPC attain a defined octagonal symmetry despite being held together by unstructured and multivalent linker NUPs? The key to this and other aspects of NPC’s structural organization may lie in the mechanism of its assembly, which remains largely enigmatic. A central hurdle is the process of membrane fusion that creates the nucleocytoplasmic conduit. How exactly is the nuclear membrane perforated? Exciting findings in budding yeast on Brl1, Apq12, and Brr6, and the enrichment of specific lipids at assembly sites suggest that we might be on the cusp of uncovering the mechanism of membrane fusion. Is this mechanism, however, entirely different in metazoa, where these proteins are not conserved? Is there a checkpoint that couples establishment of the diffusion barrier with membrane fusion?

Once assembled, NPCs are not static channels but modular machines that can fulfill a plethora of functions and adapt their protein complement and interactome in response to physiological stimuli. The currently described variations likely only scratch the surface of a multitude of NPC isoforms present in different cell types and physiological states, and it will be exciting to discover the specialized functions they fulfill. However, while NPCs are dynamic and can possibly interconvert between different variants, the core is highly stable and has to be maintained over extraordinarily long time scales. What mechanisms maintain the diffusion barrier and prevent irreversible aggregation of barrier-forming FG repeats? How are damaged NPC subunits exchanged? Are there dedicated sensors that recognize failure in NPC function and signal specific removal of defective NPCs? We are only at the beginning of the journey to understand the challenges of NPC homeostasis in long-lasting tissues and how they are connected to ageing and disease.

## Figures and Tables

**Figure 1 cells-11-01456-f001:**
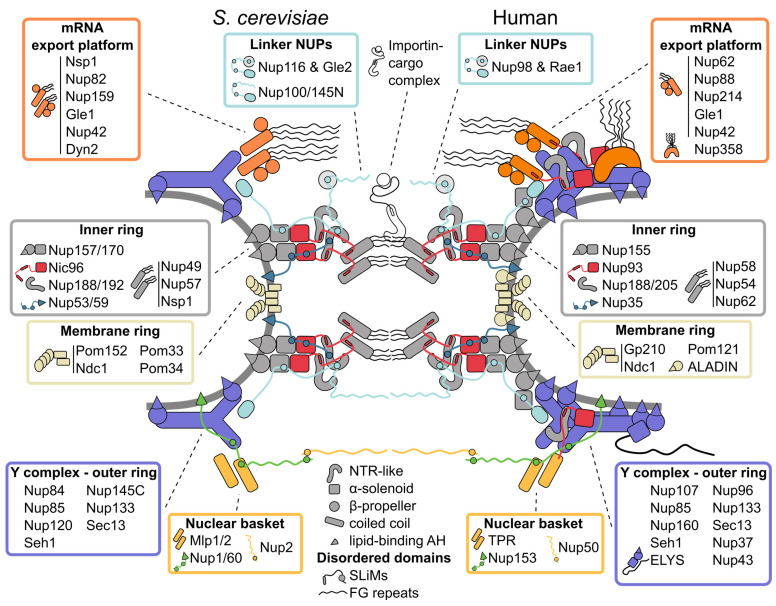
**Inventory of the budding yeast and human nuclear pore complex**. The nuclear pore complex (NPC) forms a channel connecting the nucleoplasm (bottom) with the cytoplasm (top) and is built of three concentric rings: the cytoplasmic outer ring, the inner ring, and the nucleoplasmic outer ring. The basic building blocks of the NPC are nucleoporins (NUPs), which are organized into several subcomplexes (boxes) and largely composed of only a few structural motifs (center bottom). The rigid subcomplexes are connected by disordered linkers. They contain short linear interaction motifs (SLiMs), which flexibly tie the NUPs and subcomplexes together. Multiple NUPs contain a lipid-binding amphipathic helix (AH) that helps tether the NPC to the lipid membrane. See text for details.

**Figure 2 cells-11-01456-f002:**
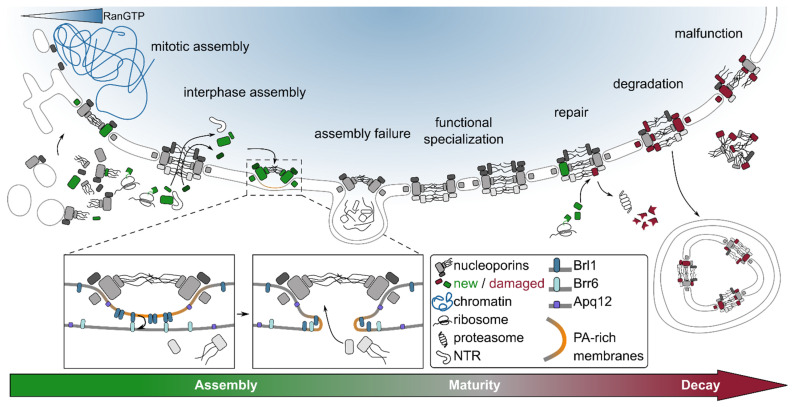
**The lifecycle of the NPC.** At the end of open mitosis, NUP recruitment to chromatin and membrane association is promoted through high local concentration of RanGTP. The NUPs seed the formation of NPCs by interacting with the re-forming NE. NPC assembly in interphase occurs “inside-out”—by inserting NPCs from the nuclear side into the sealed NE—and relies on the import of newly synthesized NUPs. It requires fusion of the inner and outer nuclear membranes by a poorly understood mechanism. The membrane fusion might involve phosphatidic acid (PA) rich membranes and the transmembrane proteins Brl1, Brr6, and Apq12 in budding yeast or torsin AAA+ ATPases in vertebrates. Cytoplasmic NUPs can join and complete NPC assembly only after successful membrane fusion. Failure in NPC assembly leads to stalled NPC intermediates (herniations) in the inner nuclear membrane enclosed by the NE and deprived of cytoplasmic NUPs. NPCs mature into compositionally and functionally different sub-populations, e.g., the budding yeast NPC can vary in the content of nuclear basket proteins or the number of nuclear Y rings. Damaged NPCs can be repaired in a “piecemeal” manner by proteasomal degradation of individual NUPs without the requirement for complete NPC disassembly. Entire NPCs can be degraded by the autophagy machinery. NPCs can accumulate damage in old age or disease, such as oxidative damage, loss of NUPs or phase-transition of FG NUPs in the cytoplasm, which leads to NPC malfunction and impaired transport. See text for details.

## Data Availability

Not applicable.

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
