# Peer review of "The Nuclear Pore Complex: Birth, Life, and Death of a Cellular Behemoth"

_cells, 2022, doi:10.3390/cells11091456_

Round 1
Reviewer 1 Report
In the manuscript “The Nuclear Pore Complex: Birth, Life and Death of a Cellular Behemoth”, Dultz and colleagues review the current understanding of the composition, biogenesis and recycling of the Nuclear Pore Complex (NPC). This is one of the most complete and insightful reviews on the nuclear pore complex in recent years, being well-balanced, comprehensive (despite the breadth of the subject matter covered) and timely. There is interesting discussion of potential evolutionary frameworks of NPC subcomplex assembly and the specificity of targeting to the nuclear membrane. The manuscript overall is clear, concise and well written with particularly edifying figures. There is little doubt this review will be well received, though there are some minor points that the authors should take into consideration before publication of the manuscript.
General minor points:
- Pom152 may well be a distant homolog of gp210, as while there is little primary sequence conservation between them, they do share strong structural homology in their repeat Ig folds. So possibly worth considering modifying the statement describing this? Interestingly, this likely homology is further underscored by the presence of gp210 in the more divergent plants – so if Pom152 and gp210 are truly unrelated then it’s a case of remarkable convergent evolution – but my money right now is that Pom152 is a divergent gp210 homolog.
- Just a point to consider - in terms of speed of interphase assembly of NPCs, yeast may also be informative in that as they do not disassemble their NEs during mitosis, all NPC assembly is presumably by the “interphase” mechanism and must be faster than “hours” as the yeast cell cycle can be as fast as 90 mins.
- The section on yeast nuclear membrane fusion (Brl1/Apq12/Brr6) seems like a departure from the flow of the rest of the review. That is to say, it seems to really dive into the details of this new body of work on these NM fusion-related proteins in a way that is a little more in depth than observed in other sections; it sticks out as a little detailed for the scope of the review and the depth of the surrounding sections. Perhaps in the interests of brevity, the authors might want to precis it a little more?
Specific minor points:
- Line 39-40: “Depending on the species, the NPC has an outer diameter of ~120-130 nm and a 39 height of 50-80 nm.” Does this necessitate a citation? Also Zimmerli, C.E et al. Science 2021 (ref. 9) suggests this outer diameter may change as a response to stress along with the inner diameter (cited below). Can we really say there is a “default” diameter of a structure that likely is underdoing significant structural rearrangements?
- Reference 7 has an incomplete author list Mosalaganti, S.; Obarska-Kosinska, A.; Siggel, M.; Turonova, B.; Zimmerli, C.E.; Buczak, K.; Schmidt, F.H.; Margiotta, E.; 804 Mackmull, M.-T.; Hagen, W. Artificial intelligence reveals nuclear pore complexity. bioRxiv 2021 should include the following information: Artificial intelligence reveals nuclear pore complexity. Shyamal Mosalaganti, Agnieszka Obarska-Kosinska, Marc Siggel, Beata Turonova, Christian E. Zimmerli, Katarzyna Buczak, Florian H. Schmidt, Erica Margiotta, Marie-Therese Mackmull, Wim Hagen, Gerhard Hummer, Martin Beck, Jan Kosinsk bioRxiv 2021.10.26.465776
- Reference 9: Zimmerli, C.E.; Allegretti, M.; Rantos, V.; Goetz, S.K.; Obarska-Kosinska, A.; Zagoriy, I.; Halavatyi, A.; Mahamid, J.; Kosinski, J.; Beck, M. Nuclear pores constrict upon energy depletion. bioRxiv 2020 should be updated to Zimmerli, C.E et al., Nuclear pores dilate and constrict in cellulo. Science, 374(6573), 2021 since the preprint has been published.
- Line 43: As the cytoplasmic ring is mentioned first, I would change the order, first the mRNA export platform and then the fishtrap-like nuclear basket and add, respectively.
- Not all the reviews (such as 11, 12, 137 and 138) are cited as “reviewed in”.
- Line 70-71: “The outer, membrane-binding layer is composed of the α-helical/ β-propeller domain paralogues Nup157/Nup170 which bind the NE via an amphipathic lipid packing sensor (ALPS) motif positioned in a loop between two β-propeller blades”. Nup157 and Nup170 are not just α-helical but α-solenoid domains; that is, they are stacked helix-turn-helix motifs, a common feature in many Nups. This is a more accurate structural description of the domains, and the authors should consider substituting this here.
- Line 91: references six conserved constituent proteins in the Y complex. Shouldn’t it be seven (Nup133, Nup120, Nup145N, Nup84, Nup85, Seh1, Sec13), as indicated in Figure 1? Or possibly clarify here a little more the compositional variation between species (as also indicated in that Figure).
- Line 166: The reference to “Vollmer, 2015” is inserted though not formatted. Likely this is a reference to current reference 108 (Vollmer, B. et al. Nup153 Recruits the Nup107-160 Complex to the Inner Nuclear Membrane for Interphasic Nuclear Pore Complex Assembly. Dev Cell). This would move this reference up to reference 67.
- Line 191: “just a mechanical gridle” should be “just a mechanical girdle”.
- Line 192: Subheading 2.5 lacks line spacing and italics.
- Line 298-299: “The NPC consists of distinct subcomplexes that take their origin on ribosomes as nascent polypeptides that must be correctly folded and bind to cognate interaction partners”. Does this require citation/attribution?
- Line 344: Previous formatting (e.g. Line 170: [reviewed in [69]]) used brackets for “reviewed in” parenthetical phrase.
- Line 358: says that “a large body of evidence suggests…” but there are no references.
- Paragraph starting with line 517: “NPC isoforms can also represent age-specific subpopulations, as exemplified by budding yeast, where a significant fraction of NPCs do not contain the basket NUPs Mlp1 and Mlp2….” The kinetic model proposed here is interesting and consistent with the stepwise assembly data cited. However, how this “aging NPC model” operates relative to the above notation that Mlp1/Mlp2 are excluded from areas of the nuclear envelope. Are there two models at play? An aging/kinetic model and a nuclear compartment model or are they intertwined?
- Line 560: They say that association of the budding yeast nuclear basket NUP Mlp1 with the NPC was found to depend on mRNA. Could authors clarify how Mlp1 association with the NPC depends on mRNA?
- Line 570: “but might in some cases involve alternative NUP complexes in the NE [223,224] ;Gozalo, 2019 #97}”. A reference Gozalo, 2019 is listed but not included in the references. Likely this reference: Gozalo, A., et al. Core Components of the Nuclear Pore Bind Distinct States of Chromatin and Contribute to Polycomb Repression Mol Cell 77(1):67-81.e7 2019.
- Lines 580-583: Shouldn’t these be regular text, not bold?
Reviewer 2 Report
Dear colleagues,
I have read the review “The Nuclear Pore Complex: Birth, Life and Death of a Cellular 2 Behemoth”, with much pleasure. The review is well written, is comprehensive and balanced and it has many interesting ideas. The field is moving fast at the moment and the review will help newcomers to understand all the new developments. Also for experts in the field the review is valuable because it clearly lays out the new questions and direction that the field can move into. At least for me, I find that several on my more disorganized ideas and thoughts were spelled out clearly by the authors and backed up with the right references. overall I find this a valuable review that I will certainly recommend to my labmembers and I fully support publication.
Some small comments:
Figure 1 is complex but correct. Just the Nup2 cartoon was less clear (make the circles bigger?); the Y complex and the Mlps having the say grey scale is a bit confusing; the cargo-kap60-95 cartoons are not explained
The reference Vollmer, 2015 166 #102 and Gozalo, 2019 #97 are not appearing as a number in the text
I would choose to write out nuclear transport machinery as the abbreviation NTM may be confusing for non-experts next to the often used NTR and NTF. Also I find CNT (for channel nucleoporin heterotrimer) confusing next to the often used NCT for nucleocytoplasmic transport.
Figure 2 is very insightful and valuable but could be made even prettier. E.g. making a perfect round vesicle with parallel membrane, always having the same distance between to NPC cartoons, having more similar membrane shapes for the assembly zoom in panels.
Author Response
"Please see the attachment."

Reviewer 3 Report
Summary:
Overall, this is a very well-written and comprehensive review that will be a highly valuable resource both for NPC experts and a wider audience. The review is very timely as well, and nicely captures the essence of several important advances in the field published over the past few years, including some novel aspects in the context of NPC assembly and mention of disease relevance. While the review could be published in its present form, several suggestions for improvement are provided below. This includes some stylistic changes to Fig. 1 and some minor additions/revisions regarding the discussion of recent literature.
Major comments:
1) Figure 1 is a bit difficult to look at. For the reader it takes some time to find a protein complex mentioned in the main text in the figure. Perhaps using more colors and match the color of the boxes (e.g. for inner ring, membrane ring, etc) with the coarse-grained/corresponding structural organization would make it easier to match name and location of the protein complexes.
2) The authors should mention/discuss a recent study from the Kirchhausen lab as it challenges the idea of an obligatory, complete mitotic NPC disassembly (10.1016/j.devcel.2021.05.015)
3) The idea of a “tight seal” (line 219) seems difficult to reconcile with transport of membrane proteins from ER to INM. Rephrasing or acknowledging this problem is recommended
4) In the context of chaperones (lines 327-339; line 649) MLF2 (ref. 126) could be mentioned as emerging candidate as well. Note also that refs. 124 and 126 reported DNAJB2 and DNAJB6 localization to NPC intermediates, respectively, this could be acknowledged.
5) On lines 177-178, the authors refer to TMEM33 as an ortholog of Pom33. To this reviewer’s knowledge, TMEM33 has not been shown to fill an orthologous role at the NPC and questions remain about its contribution to NPC structure/function. Thus, the authors should reword this sentence to reflect the open question in the field.
Minor comments:
1) Spell out species names upon first use, abbreviate after that in the following text.
2) There are a few instances of unformatted citations, see lines 166 and 570.
3) The header on line 192 should be italicized for consistency with other section headers.
Author Response
"Please see the attachment."
